# When Large Models Meet Generalized Linear Models: Hierarchy Statistical Network for Secure Federated Learning

## Abstract

Large pre-trained models perform well on many Federated Learning (FL) tasks. Recent studies have revealed that fine-tuning only the final layer of large pre-trained models can reduce computational and communication costs while maintaining high performance. We can model the final layer, which typically performs a linear transformation, as a Generalized Linear Model (GLM). GLMs offer advantages in statistical modeling, especially for anomaly detection. Leveraging these advantages, GLM-based methods can be utilized to enhance the security of the fine-tuning process for large pre-trained models. However, integrating GLMs with large pre-trained models in FL presents challenges. GLMs rely on linear decision boundaries and struggle with the complex feature representation spaces from pre-trained models. To address this, we introduce the **H**ierarchy **Stat**istical **Net**work (HStat-Net). HStat-Net refines the spaces to make them more discriminative, allowing GLMs to work effectively in FL. Based on HStat-Net, we further develop FEDRACE to detect poisoning attacks using deviance residuals from GLMs. We also provide a theorem to support FEDRACE's detection. Extensive experiments conducted on CIFAR-100, Food-101, and Tiny ImageNet demonstrate that FEDRACE significantly outperforms existing state-of-the-art defense algorithms.

## 1 Introduction

Large pre-trained models, such as CLIP Radford et al. (2021), have played a significant role in computer vision by demonstrating strong adaptability across various tasks. These models are usually fine-tuned on centralized servers, where data from multiple clients is collected to build a powerful global model. However, this centralized approach raises privacy concerns, as clients are often reluctant to share their data. Federated Learning (FL) addresses these concerns by allowing clients to train models locally without transferring raw data to a central server. Instead, only model updates are shared and aggregated, ensuring a privacy-preserving training mechanism McMahan et al. (2017); Nguyen et al. (2021). This method has proven particularly effective in privacy-sensitive fields such as healthcare and activity monitoring Yu et al. (2020); Khan et al. (2021); Cui et al. (2021).

Despite the advantages, integrating large pre-trained models into FL is challenging due to the limited computation and communication resources of mobile devices, making full model fine-tuning impractical. An efficient solution is to adapt only the final layer or the last few layers of the pre-trained model. In transfer learning, a common strategy is to replace the pre-trained model's classification head with a task-specific layer. For example, in Vision Transformer (ViT) models, the classification head is typically replaced by a linear layer to adapt to new datasets Dosovitskiy et al. (2021). Similarly, in BERT-based natural language processing tasks, both full model fine-tuning and feature-based approaches, in which a simple classification layer is added while keeping the pre-trained parameters frozen, have demonstrated effectiveness Devlin et al. (2018). In FL, adapting only the final layer reduces both computational and communication overhead while still leveraging the knowledge embedded in the pre-trained model Kornblith et al. (2019). Recent research further suggests that only training the classifier can yield good performance in FL Legate et al. (2024).

Since the final layer is typically a fully connected layer that performs a linear transformation, we can model this transformation using a Generalized Linear Model (GLM) McCullagh & Nelder (1989); Emami et al. (2020). GLMs offer several benefits in statistical modeling, particularly in anomaly

detection Hastie et al. (2009). Building on these strengths, we propose harnessing GLM-based statistical methods to protect the fine-tuning process of large pre-trained models. However, applying GLMs to the feature representation spaces generated by large pre-trained models in FL presents challenges. Since GLMs rely on linear decision boundaries, they perform best with data that is linearly separable. However, the spaces generated by large models often exhibit significant class overlap due to the complex and entangled representations learned from distributed and heterogeneous data, as discussed in Section 3.2. Therefore, we aim to disentangle these feature representation spaces to improve linear separability, allowing GLMs to work more effectively in FL. To the best of our knowledge, this is the first attempt to integrate GLMs with large models in the context of FL.

To achieve this integration, we design **H**ierarchy **Stat**istical **Net**work (HStat-Net), a novel architecture that bridges large pre-trained models with GLMs. As shown in Figure 1, HStat-Net consists of three key components[1]: a pre-trained feature extractor ($\phi$), a *Statistical Net* (**s**), and a *Task Net* (**h**). The statistical net transforms high-dimensional representations into lower-dimensional, more discriminative ones, improving class separability and enabling effective

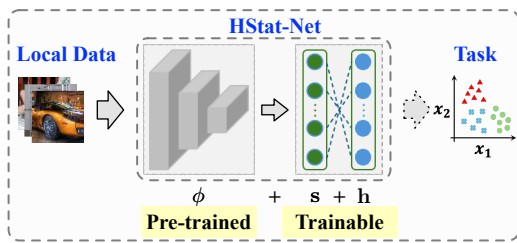

Figure 1: HStat-Net architecture: $\mathbf{w} = \mathbf{h} \circ \mathbf{s} \circ \phi$

GLM performance. The task net, functioning as a GLM, utilizes these refined representations to perform downstream tasks. Furthermore, we introduce a novel two-step training procedure for HStat-Net, which differs from traditional approaches by separately addressing the roles of **s** and **h**.

Building on the properties of HStat-Net, we develop FEDRACE, a novel mechanism for detecting poisoning attacks using deviance residuals, a statistical method employed in GLMs McCullagh & Nelder (1989). By integrating GLMs within HStat-Net, FEDRACE can accurately detect various poisoning attacks through reliable centralized evaluation. This reduces reliance on client-side validation, as seen in FLShield Kabir et al. (2024), and enhances scalability and efficiency, particularly for mobile devices. Moreover, FEDRACE does not require pre-defined parameters for the detection process, unlike FLAIR Sharma et al. (2023). Our key contributions are as follows:

- We introduce HStat-Net, a framework that bridges large pre-trained models with GLMs, facilitating the application of statistical methods and enabling new clients to quickly adapt by fine-tuning only the task-specific **h**.
- We apply HStat-Net to secure federated learning using the FEDRACE mechanism, leveraging traditional statistical methods to detect various types of poisoning attacks. Additionally, we provide a theorem to calculate the upper bound of detection error, which further supports FEDRACE's detection process.
- We conduct extensive experiments on CIFAR-100, Food-101, and Tiny ImageNet, validating that FEDRACE effectively detects various poisoning attacks while delivering outstanding performance compared to state-of-the-art defense algorithms.

## 2 BACKGROUND

### 2.1 FEDERATED LEARNING AND LARGE PRE-TRAINED MODELS

Federated learning allows a group of clients $\mathcal{N} = \{1, \ldots, N\}$ to collaboratively train a global model while preserving the privacy of local datasets. In each communication round $t$, the central server randomly selects a subset of clients $\mathcal{N}^{(t)} \subseteq \mathcal{N}$ to participate in training. The goal is to minimize the global loss function, defined as:

$$\min_{\mathbf{w}} \mathcal{L}(\mathbf{w}) = \sum_{i \in \mathcal{N}^{(t)}} \frac{|\mathcal{D}_i|}{|\mathcal{D}|} \mathcal{L}_i(\mathbf{w}), \tag{1}$$

where $\mathbf{w}$ represents the model parameters, $|\mathcal{D}_i|$ is the size of client $i$'s dataset $\mathcal{D}_i$, and $\mathcal{D} = \bigcup_{i \in \mathcal{N}^{(t)}} \mathcal{D}_i$ represents the combined local datasets.

---

[1]Pictures used as *Local Data* in Figure 1 are sourced from the Tiny ImageNet dataset Le & Yang (2015).

Large pre-trained models, such as CLIP Radford et al. (2021), enhance federated learning by accelerating convergence and improving generalizationCai et al. (2023a); Tu et al. (2024); Cai et al. (2023b). These models outperform traditional centralized training methods, even with non-IID data Tu et al. (2024). In our experiments with the CLIP model on the CIFAR-100 dataset Krizhevsky et al. (2009), we modified the model's final layer for task-specific fine-tuning. This results in $\mathbf{w} = \mathbf{h} \circ \phi$, where $\phi$ is the pre-trained part for feature extraction and $\mathbf{h}$ represents the task-specific component.

We evaluated three strategies: *Retrain* (training from scratch), *Fully Fine-Tuned* (fine-tuning the entire model), and *Partially Fine-Tuned* (training only $\mathbf{h}$ while freezing $\phi$). The results in Table 1 show that pre-trained models significantly improve test accuracy. Furthermore, training only the task-specific

Table 1: Comparisons of training methods.

| Method | Trainable Parameters | Training Time | Testing Accuracy |
|---|---|---|---|
| Retrain | $\approx 86.62M$ | 0.0423 sec | 58.32% |
| Fully | $\approx 86.62M$ | 0.0411 sec | 68.04% |
| Partially | $\approx 0.05M$ | 0.0130 sec | 75.99% |

layer $\mathbf{h}$ both increases accuracy and reduces training time per batch by 68.37%. These findings demonstrate the effectiveness of pre-trained models in federated learning and indicate that full fine-tuning may be unnecessary. The final layer $\mathbf{h}$ is similar to a *Generalized Linear Model*.

## 2.2 GENERALIZED LINEAR MODEL

Generalized linear models extend traditional linear regression by allowing the response variable to follow distributions from the exponential family, such as binomial, Poisson, or multinomial distributions. In GLMs, the expected value of the response variable is related to the linear predictors through a link function McCullagh & Nelder (1989). In GLMs, the response variable $Y$ is modeled as a linear combination of explanatory variables $\mathbf{R}$, with $\mathbf{w_h}$ representing the task-specific parameters of $\mathbf{h}$. The link function $g(\cdot)$ relates the expected value of $Y$ to the linear predictor:

$$g(\mathbb{E}[Y]) = \mathbf{R}^\top \mathbf{w_h}. \tag{2}$$

Here, $\mathbf{R}$ denotes the representations extracted by the pre-trained model, and $Y$ represents corresponding target variable, such as class probabilities. In multi-class classification, the inverse link function $g^{-1}$ is typically the softmax function, which maps linear predictors to class probabilities.

Integrating GLMs with large pre-trained models in federated learning provides advantages like flexible feature-output modeling, and enhanced anomaly detection. However, a key limitation is that GLMs rely on linear decision boundaries, making them less effective in capturing complex, non-linear relationships embedded in the data. When class separation in the feature representation space is unclear, GLMs may struggle with outlier detection, leading to biased model parameters and distortion towards extreme values Montgomery et al. (2021).

## 3 HIERARCHY STATISTICAL NETWORK

We introduce a **H**ierarchy **Stat**istical **Net**work (HStat-Net) to integrate large pre-trained models with GLMs. This architecture enables efficient and secure fine-tuning for large pre-trained models, as shown in Figure 1. HStat-Net consists of three key components:

**Pre-trained Feature Extractor** ($\phi$) The first component, the pre-trained feature extractor $\phi$, processes raw input data $\mathbf{x}$ into an initial feature representation vector: $\mathbf{z} = \phi(\mathbf{x})$. During federated learning, the parameters of $\phi$ remain frozen to reduce computational costs.

**Statistical Net** ($\mathbf{s}$) The second component, the statistical net $\mathbf{s}$, refines the representation vector $\mathbf{z}$ into a more discriminative representation: $\mathbf{r} = \mathbf{s}(\mathbf{z})$. Here, $\mathbf{r}$ serves as the explanatory variables $\mathbf{R}$ in Equation 2.

**Task Net** ($\mathbf{h}$) The final component, the Task Net $\mathbf{h}$, performs the downstream task, such as classification, using the representation $\mathbf{r}$: $\hat{y} = \mathbf{h}(\mathbf{r})$. Thus, for client $i$, the complete HStat-Net is:

$$\hat{y}_i = \mathbf{h}_i(\mathbf{s}_i(\phi_i(\mathbf{x}))) = \psi_i(\phi(\mathbf{x})), \tag{3}$$

where $\psi_i$ denotes the combined operations of $\mathbf{h}_i$ and $\mathbf{s}_i$. To reduce model complexity and enhance privacy, the statistical net $\mathbf{s}$ performs dimensionality reduction from $\mathbf{z} \in \mathbb{R}^D$ to $\mathbf{r} \in \mathbb{R}^d$, where $d \ll D$. During the training process, only $\mathbf{h}$ and $\mathbf{s}$ are trained locally. Clients send their trained $\mathbf{h}_i$ and $\mathbf{s}_i$ to the central server, where they are aggregated with equal weights to obtain the globally updated $\psi = \mathbf{h} \circ \mathbf{s}$, following a process similar to FedAvg McMahan et al. (2017).

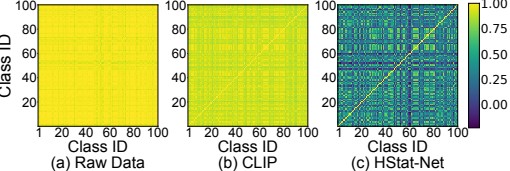

Figure 2: Two-step training process.

Figure 3: Class-wise cosine similarity.

## 3.1 TRAINING THE HSTAT-NET

To enable HStat-Net to effectively refine the representation space, we apply *Triplet Loss* Schroff et al. (2015); Wen et al. (2016) to the statistical net (**s**). This loss improves the discrimination by bringing similar samples closer and pushing dissimilar ones farther apart, thereby creating feature representation spaces more suitable for GLMs. The *Triplet Loss* is defined as:

$$\mathcal{L}_{\text{Triplet}} = \sum_{l \in \mathcal{D}^{\text{batch}}} \max(||\mathbf{r}_l - \mathbf{r}_l^p||2^2 - ||\mathbf{r}_l - \mathbf{r}_l^n||2^2 + 0.1, 0), \tag{4}$$

where $\mathbf{r}_l$ is the representation of the current sample (anchor) refined by **s**, $\mathbf{r}_l^p$ is the representation of another sample from the same class (positive), and $\mathbf{r}_l^n$ is from a different class (negative). For the task net (**h**), we use the *Cross-Entropy* loss:

$$\mathcal{L}_{\text{CE}} = - \sum_{l \in \mathcal{D}^{\text{batch}}} \sum_{c=1}^{C} y_l^c \log \hat{y}_l^c, \tag{5}$$

where $C$ is the number of classes, $y_l^c$ the one-hot encoded true label, and $\hat{y}_l^c$ the predicted probability for class $c$. The statistical net (**s**) refines representations, while the task net (**h**) functions as a GLM. To optimize both components without gradient conflicts in round $t$, where gradients from different loss functions may be in opposing directions, client $i$ follows a two-step training process based on the previously aggregated $\mathbf{h}^{t-1} \circ \mathbf{s}^{t-1}$, as shown in Figure 2:

① **Train Task Net ($\mathbf{h}_i^t$):** Freeze the statistical net and minimize $\mathcal{L}_{\text{CE}}$ using local data.

② **Train Statistical Net ($\mathbf{s}_i^t$):** Freeze the task net and minimize $\mathcal{L}_{\text{Triplet}}$ using local data.

This training strategy allows each component to specialize in its own task. During local training, local task nets are updated with $\mathbf{s}^{t-1}$ frozen, ensuring consistent refined representations across all clients, which mitigates data heterogeneity, including for new clients. Restarting the entire training process for new clients is impractical due to high training costs Li et al. (2020), but with a global statistical network **s** established, new clients can fine-tune only their task net ($\mathbf{h}_i$), reducing extensive retraining needs. This strategy enhances the efficiency and generalization of federated learning, especially in environments with heterogeneous data and dynamic client participation.

## 3.2 EXPERIMENTAL VALIDATIONS

Similar to Section 2, we conducted experiments on the CIFAR-100 dataset, using CLIP as $\phi$ in HStat-Net. The dataset was distributed across 64 clients, with samples assigned following a Dirichlet distribution $\text{Dir}_N(\alpha)$ to simulate a realistic FL environment. The parameter $\alpha$ controls the degree of non-IID data distribution, where smaller values represent higher heterogeneity. We chose a moderate value of $\alpha = 0.5$, consistent with previous studies Oh et al. (2022); Dai et al. (2022); Chen & Chao (2022); Fang et al. (2020); Wang et al. (2020b;c); Jiang et al. (2022).

**Representation analysis** Figure 3 illustrates the class-wise similarities for raw data, representations extracted by $\phi$, and refined representations from HStat-Net (**s**), using cosine similarity for comparison across 100 classes. The raw data shows a high similarity, averaging 0.976, and while CLIP significantly enhances downstream tasks, it only reduces the overall similarity to 0.908. In contrast, HStat-Net reduces class-wise similarity more substantially, achieving an average of 0.339.

We further evaluated class separability using Fisher's Criterion and Mutual Information (MI) Fisher (1936); Dhir & Lee (2008); Estévez et al. (2009). Fisher's Criterion measures the ratio of between-class to within-class variance, with higher scores

Table 2: Representation analysis.

| - | Raw | CLIP | HStat-Net |
|---|---|---|---|
| Fisher | 0.149 | 0.480 | **1.602** |
| MI | 0.162 | 0.275 | **0.556** |

indicating better class separation. MI quantifies the dependency between representations and class labels, capturing the extent to which representations convey information about class distinctions. As shown in Table2, HStat-Net significantly improves class separability, with a 3.34x increase in the Fisher score and a 2.02x improvement in MI compared to CLIP. These results highlight HStat-Net's effectiveness in enhancing class separation, facilitating the application of GLMs in federated learning.

**Generalization analysis** With HStat-Net's representation enhancement, experiments show a 1.19% improvement in overall model accuracy. However, generalization capability is crucial for applying HStat-Net in real-world scenarios. To assess this, we trained an FL model using HStat-Net with $\alpha = 0.5$, then introduced 10 new clients not involved in the original training, using $\alpha = \{0.1, 0.5, 0.9\}$. Each new client underwent one epoch of fine-tuning, and we calculated the average accuracy across these clients. For consistency, their test datasets were partitioned using the same $\alpha$ values.

We evaluated two scenarios for the new clients: (i) the model follows the traditional CLIP approach, represented as $\mathbf{w}_{\text{new}} = \mathbf{h} \circ \phi$, and (ii) the model uses HStat-Net, represented as $\mathbf{w}_{\text{new}} = \mathbf{h} \circ \mathbf{s} \circ \phi$. Since both $\phi$ and $\mathbf{s}$ are pre-trained, only the task net ($\mathbf{h}$) requires fine-tuning for new clients. As shown in Table 3, HStat-Net consistently outperforms the traditional model in managing new clients.

Table 3: Generalization analysis.

| - | $\alpha = 0.1$ | $\alpha = 0.5$ | $\alpha = 0.9$ |
|---|---|---|---|
| CLIP | 23.78% | 12.90% | 9.67% |
| HStat-Net | **66.85%** | **55.64%** | **52.60%** |

## 4 HStat-Net for Secure Federated Learning

### 4.1 Attacks in Federated Learning

Federated learning is vulnerable to various poisoning attacks, which can severely degrade the global model's performance. These attacks are typically classified based on the adversary's intent:

- **Untargeted attacks** Baruch et al. (2019); Fang et al. (2020); Xie et al. (2020); Shejwalkar & Houmansadr (2021): The adversary seeks to degrade the overall performance of the global model, reducing its accuracy (ACC) across all inputs. ACC is defined as:

$$\text{ACC} = \mathbb{E}_{(\mathbf{x},y)\sim\mathcal{D}}[\mathbb{P}(\mathbf{G}(\mathbf{x}) = y)], \tag{6}$$

where $\mathbf{x}$ is the input, $y$ is the true label, and $\mathbf{G}(\mathbf{x})$ is the global model's prediction.

- **Targeted attacks** Bhagoji et al. (2019); Sun et al. (2019); Chen et al. (2021): Here, the adversary focuses on specific samples or classes, aiming to reduce the model's accuracy for those targets. The effect of the attack is measured by the attack success rate (ASR):

$$\text{ASR} = \mathbb{E}_{(\mathbf{x},y)\sim\mathcal{D}, y=y^{\text{attack}}}[\mathbb{P}(\mathbf{G}(\mathbf{x}) = y^{\text{target}})]. \tag{7}$$

- **Backdoor attacks** Bagdasaryan et al. (2020); Wang et al. (2020a); Xie et al. (2019); Nguyen et al. (2022); Bagdasaryan & Shmatikov (2021): The adversary introduces a hidden malicious sub-task, causing the model to misclassify inputs containing specific triggers into a designated class $y^{\text{target}}$. The effectiveness is measured by backdoor accuracy (BA):

$$\text{BA} = \mathbb{E}_{(\mathbf{x},y)\sim\mathcal{D}, y=y^{\text{attack}}}[\mathbb{P}(\mathbf{G}(\mathbf{T}_{ri}(\mathbf{x})) = y^{\text{target}})], \tag{8}$$

where $\mathbf{T}_{ri}(\cdot)$ is the trigger function that embeds the backdoor into the input $\mathbf{x}$.

**Adversary's capability** The adversary can control up to $M$ out of $N$ clients, with $M/N < 50\%$, as exceeding this threshold would undermine Byzantine-robust aggregation rules Fang et al. (2020); Shejwalkar & Houmansadr (2021). In each communication round, the central server randomly selects a subset of clients $\mathcal{N}^{(t)}$, and the number of malicious clients $m$ in this subset can vary. The adversary has full access to the global model parameters and can execute attacks through the clients under their control Xie et al. (2018); Bhagoji et al. (2019); Bagdasaryan et al. (2020); He et al. (2020).

**Adversary's knowledge** The adversary operates with limited knowledge, unaware of the aggregation rule or gradient updates from benign clients. They only have access to information about the malicious clients under their control, making the scenario more realistic.

Despite the robustness of pre-trained large models, traditional poisoning attacks can still be highly effective, even when only task-specific layers are fine-tuned Kurita et al. (2020); Li et al. (2021). When the parameters of the pre-trained model $\phi$ are frozen and only $\mathbf{h}$ is updated, attackers can exploit

this to manipulate model behavior without substantially affecting the overall model. Our experiments on CIFAR-100 show that an untargeted attack Xie et al. (2020) reduces the model's accuracy from 75.99% to 64.24%. To counter these attacks in federated learning, we propose FEDRACE (**Fed**erated **R**epresentation-based **A**daptive **C**lient **E**valuation), which leverages the properties of GLMs in HStat-Net to detect and remove them during global aggregation of $\psi = \mathbf{h} \circ \mathbf{s}$ on the central server.

## 4.2 OBTAINING RELIABLE GLOBAL REPRESENTATIONS

Detecting malicious clients in FL is challenging because of data heterogeneity and restricted visibility into local computations. To address this, all clients use a globally synchronized statistical net ($\mathbf{s}$) when training task nets, to ensure consistent class-wise representations across all clients.

As described in Section 3, HStat-Net increases intra-class similarity and reduces inter-class similarity, thereby making class-wise representations an effective approach for detecting attacks. When client $i$ performs task net training in round $t$, it simultaneously computes the representations:

$$\mathbf{r}_i^c = \frac{1}{n_i^c} \sum_{l=1}^{n_i^c} \mathbf{r}_{i,l}^c, \text{ where } \mathbf{r}_{i,l}^c = \mathbf{s}^{t-1}(\mathbf{z}_{i,l}^c), \tag{9}$$

where $\mathbf{r}_{i,l}^c$ represents the $l$-th sample of class $c$ on client $i$, and $n_i^c$ is the number of samples for class $c$. The server aggregates these representations to form global class representations:

$$\mathbf{r}_{\text{global}}^c = \text{median}(\mathbf{r}_i^c | i \in \mathcal{N}^{(t)}). \tag{10}$$

Here, the *median* enhances robustness against outliers by reducing the influence of extreme values Yin et al. (2018), ensuring a reliable global representation for detecting malicious activity.

HStat-Net's two-step training process ensures that the representations extracted using $\mathbf{s}^{t-1}(\cdot)$ are aligned with the current task net $\mathbf{h}_i^t$, maintaining consistency across clients. Moreover, by sending the representations extracted by $\mathbf{s}^{t-1}$, current $\mathbf{s}^t$ and $\mathbf{h}^t$ to the server, privacy is enhanced as reconstructing the original data becomes significantly more difficult. By modeling the task net as a generalized linear model, we establish a statistical framework that compares client models to the expected prediction of corresponding global representation, enabling the detection of malicious clients.

## 4.3 REPRESENTATION-BASED DETECTION

Once reliable representations are obtained, we use *Deviance Residuals* McCullagh & Nelder (1989) to detect malicious clients by evaluating how well each client's $\mathbf{h}$ aligns with the expected prediction. The deviance residual for client $i$ on class $c$ can be computed as (details in Appendix A.1):

$$\Delta_i^c = -2 \log(\hat{y}_{\mathbf{r}}^c), \tag{11}$$

where $\hat{y}_{\mathbf{r}}^c = \mathbf{h}_i(\mathbf{r}_{\text{global}}^c)$ is the predicted probability for class $c$ based on the global representation. High residuals indicate significant deviations from expected prediction, which may suggest malicious activity. To detect a wide range of poisoning attacks, including class-specific attacks, we aggregate class-level residuals into a client-level residual:

$$\Delta_i = \sum_{c=1}^{C} \Delta_i^c \times \log(\Delta_i^c). \tag{12}$$

This formulation emphasizes classes with higher residuals, helping to identify clients who manipulate specific classes while behaving normally in others. Once the residuals $\{\Delta_i\}_{i \in \mathcal{N}^{(t)}}$ are computed, they are sorted in ascending order:

$$\Delta_{[1]} \leq \Delta_{[2]} \leq \cdots \leq \Delta_{[n]}, \tag{13}$$

where $n = |\mathcal{N}^{(t)}|$ is the number of selected clients in round $t$. Typically, the residuals follow below:

$$\Delta_{[i]} = \begin{cases} O_p(1), & \text{if } i \leq p \\ A_{\mathcal{M}} + O_p(1), & \text{otherwise} \end{cases} \tag{14}$$

where $A_{\mathcal{M}} > 0$ is a constant tied to the poisoning attack, and $O_p(1)$ represents a term bounded in probability. Since estimating $A_{\mathcal{M}}$ is difficult without prior knowledge of the attack, we analyze the residual property to determine a cutoff for separating benign and malicious clients.

**Theorem 1.** *Assuming the expected deviance residuals of benign clients $\mu_\mathcal{B}$ and malicious clients $\mu_\mathcal{M}$ satisfy $\mu_\mathcal{B} < \mu_\mathcal{M}$, and their variances are bounded by a constant $\sigma^2$, the total misclassification rate (TMR) of the detection algorithm is bounded by (proof in Appendix A.2):*

$$TMR \leq \frac{4\sigma^2}{(\mu_\mathcal{M} - \mu_\mathcal{B})^2}. \tag{15}$$

Based on Theorem 1, we determine a threshold $\hat{p}$ that separates clients into two groups. Clients with residuals exceeding $\Delta_{[\hat{p}]}$ are identified as potentially malicious. For each value between 1 and $n$, we calculate an upper bound and select the point that minimizes the bound as the final $\hat{p}$.

To improve detection accuracy, we implement majority voting across multiple iterations. In each iteration, a random subset $\mathcal{N}_{\text{sub}}^{(t)} \subset \mathcal{N}^{(t)}$ is selected to compute global representations and carry out detection. A client is given a vote if classified as malicious in that iteration. After $K$ iterations, the total amount of votes for each client is calculated as:

$$\text{Votes}_i = \sum_{k=1}^{K} \mathbb{I}_{\Delta_i^{(k)} > \Delta_{[\hat{p}_k]}}, \tag{16}$$

where $\mathbb{I}_.$ is the indicator function, and $\Delta_i^{(k)}$ is the deviance residual for client $i$ in iteration $k$. Finally, a client is classified as malicious if it receives votes in more than half of the iterations ($K/2$).

## 5 EXPERIMENTS

### 5.1 EXPERIMENTAL SETUP

**Dataset and models**. We conducted experiments on three widely adopted computer vision datasets: CIFAR-100, Food-101, and Tiny ImageNet. CIFAR-100 Krizhevsky et al. (2009) contains 100 classes of natural images, with 600 images per class, and is widely used for multi-class classification tasks. Food-101 Bossard et al. (2014) consists of 101 categories of food images, commonly used for fine-grained visual classification. Tiny ImageNet Le & Yang (2015) is a smaller version of ImageNet, with 200 classes and 500 images per class, and is often used for benchmarking large-scale image classification tasks. We use the CLIP model without its final layer, as the feature extractor ($\phi$) in HStat-Net. Both $\mathbf{s}$ and $\mathbf{h}$ are single fully-connected layers, with $d = 256$ for $\mathbf{r} \in \mathbb{R}^d$. To evaluate FEDRACE, we use the same HStat-Net architecture across all defense methods and attack baselines. We also test the scalability of HStat-Net by replacing $\phi$ with ResNet-152 He et al. (2016).

**Baselines**. We compare FEDRACE to several defense methods, including FLShield Kabir et al. (2024), FedRoLA Yan et al. (2024), FLAIR Sharma et al. (2023), Trimmed-mean Yin et al. (2018); Xie et al. (2018), and Multi-Krum Blanchard et al. (2017). We also assess it against poisoning attacks: (i) untargeted attacks such as the Min-Max Shejwalkar & Houmansadr (2021) and Inner Product Manipulation Attack (IPMA) Xie et al. (2020); and (ii) targeted attacks, including the Targeted Label Flipping Attack (TLFA) Tolpegin et al. (2020) and two backdoor attacks: Edge-case Backdoor Attack (ECBA) Wang et al. (2020a) and Distributed Backdoor Attack (DBA) Xie et al. (2019).

**Parameter settings**. All experiments were run on NVIDIA RTX A4500 GPUs. Each experiment was repeated with four random seeds, and the standard deviation is reported. We simulate a FL setup with 64 clients ($N = 64$), including 16 malicious clients ($M = 16$), with 16 clients randomly selected in each training round ($n = 16$). Detection iterations in FEDRACE are set to $K = \lceil \frac{n}{2} \rceil$, with $|\mathcal{N}_{\text{sub}}^{(t)}| = \lceil \frac{n}{2} \rceil$. The data distribution follows a Dirichlet distribution with $\alpha = 0.5$. Local training runs for three epochs with a learning rate of 0.001 and a batch size of 128 across all datasets.

### 5.2 EXPERIMENTAL RESULTS

We focus on evaluating FEDRACE, which integrates HStat-Net into secure federated learning.

**Main results** Table 4 summarizes the performance of various defense methods against five types of poisoning attacks, focusing on accuracy (ACC), attack success rate (ASR), and backdoor accuracy (BA). The results highlight significant differences across the defense methods. Notably, the FEDRACE algorithm consistently delivers strong performance across all datasets and attack types.

Table 4: Comparisons between FEDRACE and state of the arts (%).

| Dataset | Defense | Untargeted | | Targeted | | | | | |
|---|---|---|---|---|---|---|---|---|---|
| | | Min-Max | IPMA | TLFA | | ECBA | | DBA | |
| | | ACC | ACC | ASR | ACC | BA | ACC | BA | ACC |
| CIFAR-100 | Multi-krum | $72.59_{0.27}$ | $76.16_{0.32}$ | $1.52_{0.10}$ | $75.93_{0.28}$ | $20.05_{0.11}$ | $76.03_{0.31}$ | $23.20_{0.28}$ | $75.68_{0.27}$ |
| | Trimmed-mean | $75.15_{0.35}$ | $76.43_{0.27}$ | $1.79_{0.25}$ | $75.83_{0.24}$ | $10.34_{0.26}$ | $76.53_{0.26}$ | $12.16_{0.29}$ | $76.65_{0.26}$ |
| | FLAIR | $73.07_{0.29}$ | $75.74_{0.27}$ | $0.61_{0.16}$ | $74.49_{0.30}$ | $1.30_{0.23}$ | $76.21_{0.32}$ | $0.96_{0.17}$ | $75.65_{0.28}$ |
| | FedRoLA | $76.05_{0.33}$ | $76.84_{0.28}$ | $11.92_{0.28}$ | $74.88_{0.29}$ | $39.28_{0.28}$ | $76.47_{0.30}$ | $2.89_{0.28}$ | $77.04_{0.27}$ |
| | FLShield | $76.86_{0.24}$ | $76.66_{0.25}$ | $2.27_{0.29}$ | $75.63_{0.28}$ | $1.67_{0.28}$ | $76.81_{0.27}$ | $1.46_{0.27}$ | $76.99_{0.31}$ |
| | FEDRACE | $76.69_{0.32}$ | $76.99_{0.32}$ | $0.07_{0.10}$ | $77.02_{0.33}$ | $0.06_{0.11}$ | $76.98_{0.31}$ | $0.36_{0.23}$ | $77.21_{0.31}$ |
| Food-101 | Multi-krum | $52.31_{0.33}$ | $55.70_{0.27}$ | $2.07_{0.13}$ | $55.85_{0.27}$ | $20.22_{0.13}$ | $55.87_{0.28}$ | $49.13_{0.30}$ | $55.23_{0.29}$ |
| | Trimmed-mean | $54.37_{0.31}$ | $56.37_{0.31}$ | $2.34_{0.26}$ | $56.08_{0.28}$ | $27.58_{0.29}$ | $56.22_{0.32}$ | $30.84_{0.28}$ | $56.54_{0.29}$ |
| | FLAIR | $53.16_{0.30}$ | $54.27_{0.25}$ | $0.43_{0.15}$ | $52.09_{0.29}$ | $5.67_{0.30}$ | $55.24_{0.29}$ | $1.48_{0.25}$ | $53.33_{0.29}$ |
| | FedRoLA | $56.40_{0.27}$ | $55.59_{0.29}$ | $12.74_{0.29}$ | $54.10_{0.29}$ | $45.27_{0.26}$ | $56.16_{0.31}$ | $8.14_{0.28}$ | $56.51_{0.28}$ |
| | FLShield | $56.24_{0.29}$ | $56.07_{0.31}$ | $14.02_{0.32}$ | $54.76_{0.30}$ | $6.36_{0.29}$ | $56.25_{0.31}$ | $1.44_{0.28}$ | $56.65_{0.27}$ |
| | FEDRACE | $56.38_{0.27}$ | $56.76_{0.26}$ | $0.27_{0.16}$ | $56.68_{0.27}$ | $0.31_{0.16}$ | $56.70_{0.26}$ | $1.01_{0.31}$ | $56.72_{0.27}$ |
| Tiny ImageNet | Multi-krum | $71.04_{0.26}$ | $72.38_{0.26}$ | $0.63_{0.10}$ | $72.70_{0.27}$ | $19.27_{0.12}$ | $72.85_{0.27}$ | $45.71_{0.29}$ | $72.05_{0.26}$ |
| | Trimmed-mean | $71.95_{0.28}$ | $72.44_{0.29}$ | $0.95_{0.22}$ | $72.74_{0.28}$ | $33.06_{0.28}$ | $72.33_{0.30}$ | $35.09_{0.23}$ | $72.67_{0.25}$ |
| | FLAIR | $71.23_{0.35}$ | $72.59_{0.28}$ | $0.28_{0.19}$ | $70.58_{0.28}$ | $4.43_{0.28}$ | $71.89_{0.28}$ | $0.24_{0.15}$ | $70.91_{0.30}$ |
| | FedRoLA | $73.36_{0.21}$ | $72.78_{0.29}$ | $4.87_{0.27}$ | $71.92_{0.29}$ | $47.14_{0.28}$ | $72.73_{0.25}$ | $4.75_{0.28}$ | $73.13_{0.21}$ |
| | FLShield | $73.29_{0.24}$ | $73.19_{0.32}$ | $9.85_{0.28}$ | $71.84_{0.29}$ | $5.84_{0.28}$ | $73.11_{0.28}$ | $0.53_{0.19}$ | $73.21_{0.32}$ |
| | FEDRACE | $73.06_{0.29}$ | $73.40_{0.29}$ | $0.07_{0.10}$ | $73.24_{0.31}$ | $0.08_{0.10}$ | $73.44_{0.29}$ | $0.13_{0.13}$ | $73.42_{0.29}$ |

Specifically, FLAIR demonstrates effectiveness in detecting both untargeted and targeted attacks. For example, on CIFAR-100, FLAIR achieves an ASR of 0.61% and a BA of 1.30%, with similar trends across other datasets. However, its heavy reliance on prior knowledge of the number of malicious clients leads to higher false positive rates across different attack scenarios. FedRoLA improves ACC during untargeted attacks by analyzing client model similarity, but it struggles under targeted and backdoor attacks, with significantly higher ASR and BA. For instance, under TLFA on CIFAR-100, FedRoLA records an ASR of 11.92%, while its BA reaches 39.28% under ECBA. In contrast, FLShield maintains lower ASR and BA across all attack types, such as 2.27% ASR and 1.67% BA under TLFA and ECBA on CIFAR-100, though it slightly underperforms FedRoLA in ACC in certain cases. Other traditional methods, like Trimmed-mean and Multi-Krum, show varied performance across datasets. By comparison, the FEDRACE algorithm effectively detects all poisoning attacks using deviance residuals, achieving the lowest ASR and BA while maintaining high ACC across all datasets and attack types. For example, on CIFAR-100, FEDRACE achieves an ACC of 76.69% under untargeted attacks, with an ASR of only 0.07% and a BA of 0.06% under TLFA and ECBA. On Food-101 and Tiny ImageNet, FEDRACE similarly demonstrates near-zero ASR and BA, with ACC rates of 56.72% and 73.42% under DBA attacks, respectively. These results highlight that FEDRACE not only surpasses other defense methods in attack detection but also maintains strong overall model performance, ensuring stability and efficiency across diverse application scenarios.

**Evaluations on cutoff point in FEDRACE** Here, we further evaluate the accuracy of the estimated cutoff point for detecting malicious clients in FEDRACE. Theorem 1 ensures that the estimated cutoff $\hat{p}$ closely approximates the true cutoff $p^*$. We measure this accuracy by calculating $|\hat{p} - p^*|$ across different datasets and poisoning attacks. A smaller difference indicates better alignment between $\hat{p}$ and $p^*$, as shown in Table 5.

Table 5: Evaluations on $\hat{p}$

| $|\hat{p} - p^*|$ | CIFAR-100 | Food-101 | ImageNet |
|---|---|---|---|
| Min-Max | $0.11_{0.26}$ | $0.10_{0.26}$ | $0.17_{0.24}$ |
| IPMA | $0.03_{0.12}$ | $0.04_{0.14}$ | $0.03_{0.16}$ |
| TLFA | $0.04_{0.20}$ | $0.04_{0.18}$ | $0.03_{0.13}$ |
| ECBA | $0.02_{0.12}$ | $0.03_{0.14}$ | $0.02_{0.11}$ |
| DBA | $0.15_{0.28}$ | $0.06_{0.16}$ | $0.19_{0.25}$ |

The results show that attack methods like ECBA produce the smallest estimation errors across all datasets, with a minimum of 0.02 on Tiny ImageNet, indicating high accuracy. IPMA and TLFA also show small errors, particularly on CIFAR-100 and Food-101. In contrast, DBA results in larger errors, especially on CIFAR-100 and Tiny ImageNet, indicating lower accuracy under this attack. The standard deviations suggest that methods like DBA and Min-Max exhibit greater variability, also indicating that FEDRACE may be less stable in certain attack scenarios.

**Evaluations on detection accuracy in FEDRACE** We evaluate the effectiveness of FEDRACE by measuring its True Positive Rate (TPR) and False Positive Rate (FPR). TPR indicates the proportion of correctly identified malicious clients, while FPR reflects the misclassification rate of benign clients. As shown in Figure 4, FEDRACE consistently achieves high detection accuracy, with TPR values exceeding 0.97 across all datasets. For example, under the Min-Max attack, FEDRACE

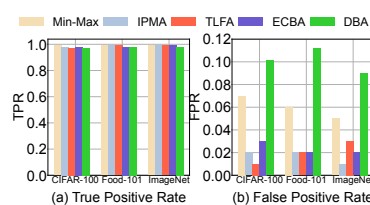

Figure 4: TPR/FPR in FEDRACE

maintains a TPR of 0.99 on CIFAR-100, Food-101, and Tiny ImageNet, demonstrating its strong ability to detect malicious clients. FEDRACE also performs effectively against the IPMA, TLFA, and ECBA attacks, with TPR values ranging from 0.97 to 0.99, further illustrating its robust defense.

In terms of FPR, FEDRACE shows low false positive rates, particularly against the IPMA and TLFA attacks, with an FPR as low as 0.01 on CIFAR-100. This demonstrates FEDRACE's ability to effectively minimize the misclassification of benign clients. However, the FPR is slightly higher under DBA attacks, ranging from 0.09 to 0.11 across datasets, indicating a higher likelihood of false positives in these cases. FEDRACE may exhibit increased error rates in certain attack scenarios.

**FEDRACE with ResNet-152** In previous results, we used $\phi$ = CLIP; here, we replace CLIP with ResNet-152 to assess FEDRACE's performance with a different feature extractor. As different $\phi$ models can affect accuracy, we continue using TPR and FPR for evaluation. As shown in Figure 5, FEDRACE maintains strong performance with ResNet-152, achieving TPR/FPR values similar to those seen with CLIP in Figure 4. This confirms that HStat-Net effectively bridges the gap between large models and GLMs, regardless of the choice of feature extractor.

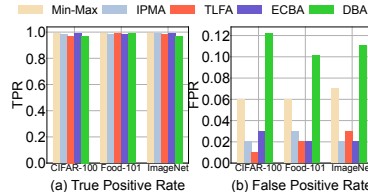

Figure 5: TPR/FPR on ResNet-152

**FEDRACE under different non-IID cases** Previous results focus on $\alpha = 0.5$. Here, we evaluate FEDRACE under $\alpha = 0.1$ (extreme non-IID) and $\alpha = 0.9$ (near IID) on the Tiny ImageNet dataset. As shown in Figure 6, FEDRACE performs well across all scenarios, with FPR decreasing as $\alpha$ increases. This robustness stems from the class-level representation-based detection, which remains effective despite variations in data distributions across federated learning scenarios.

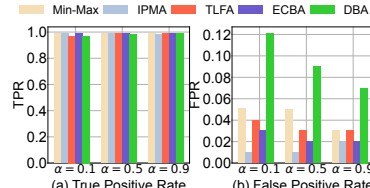

Figure 6: TPR/FPR on $\alpha$

**FEDRACE under different value of $M$** We also investigate how different amounts of malicious clients affect detection performance. To evaluate this, we test FEDRACE with $M = 8$ and $M = 24$ out of 64 total clients. As shown in Figure 7, when $M = 8$, FPR decreases compared to $M = 16$, meaning FEDRACE is less likely to misclassify benign clients as malicious. When $M$ increases to 24, FPR rises slightly, but overall detection performance remains stable. This indicates that the detection mechanism remains effective under varying levels of attack intensity.

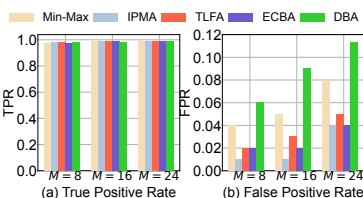

Figure 7: TPR/FPR on $M$

## 6 RELATED WORK

Models like ResNet-152 and DenseNet addressed the vanishing gradient problem and improved feature propagation, significantly boosting performance He et al. (2016); Huang et al. (2017). CLIP further advanced multi-modal learning by combining visual and textual data Radford et al. (2021). However, deploying large models poses challenges related to computational cost and privacy concerns Liu et al. (2019); Xu et al. (2021). Federated learning offers a decentralized training solution, but integrating large models remains difficult due to communication overhead and data heterogeneity McMahan et al. (2017); Li et al. (2020). Despite these advances, limited research has explored integrating GLMs with large models. While techniques like GAMs and LIME are useful, they are not specifically designed for large models in FL Caruana et al. (2015); Ribeiro et al. (2016).

## 7 CONCLUSIONS AND LIMITATIONS

We propose HStat-Net, an architecture trained using a novel two-step method, which bridges large pre-trained models with GLMs. Building on HStat-Net, we develop FEDRACE, which leverages GLM-based statistical methods to detect poisoning attacks in federated learning. However, HStat-Net is currently tailored for classification tasks in FL, posing challenges for extending to other tasks, such as text generation. Additionally, the refined representation space is not ideally linearly separable, resulting in higher false positive rates that require further improvement.

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

# A APPENDIX

## A.1 DETAILS FOR EQUATION 11

In this work, we treat the task net $\mathbf{h}$ in HStat-Net as a Generalized Linear Model, where feature representations refined by the statistical net $\mathbf{s}$ are linearly transformed and passed through a softmax function to produce class probabilities. The linearly separable representation space generated by statistical net $\mathbf{s}$ renders the task net equivalent to a multinomial logistic regression. Following previous works McCullagh & Nelder (1989); Agresti (2013); Dobson & Barnett (2018), we analyze model fit using deviance residuals, which quantify the model's alignment with expected prediction.

For each aggregated global representation $\mathbf{r}_{\text{global}}^c$ of class $c$, the true class label is denoted by the indicator variable $y_{\mathbf{r}}^c$:

$$y_{\mathbf{r}}^c = \begin{cases} 1, & \text{if global representation belongs to class } c; \\ 0, & \text{otherwise.} \end{cases} \tag{17}$$

The task net trained on client $i$ predicts the probability that representation $\mathbf{r}_{\text{global}}^c$ belongs to class $c$, denoted by $\hat{y}_{\mathbf{r}}^c$. The log-likelihood function of the trained task net for representation $\mathbf{r}_{\text{global}}^c$ is:

$$L_i^c = \sum_{c=1}^{C} y_{\mathbf{r}}^c \log(\hat{y}_{\mathbf{r}}^c) = \log(\hat{y}_{\mathbf{r}}^c), \tag{18}$$

which resembles the log-likelihood of multinomial logistic regression in the GLM framework. Assume a saturated model that perfectly fits the representation, where the predicted probabilities match the observed labels, which are one-hot encoded:

$$\hat{y}_{\mathbf{r},\text{saturated}}^c = y_{\mathbf{r}}^c, \tag{19}$$

Then, the log-likelihood for the saturated model is:

$$L_{\text{saturated}}^c = \sum_{c=1}^{C} y_{\mathbf{r}}^c \log(\hat{y}_{\mathbf{r},\text{saturated}}^c) = \log(\hat{y}_{\mathbf{r},\text{saturated}}^c), \tag{20}$$

Since $y_{\mathbf{r}}^c$ is either 0 or 1, and $\log(1) = 0$, only the terms where $y_{\mathbf{r}}^c = 1$ contribute, leading to a log-likelihood of zero for the saturated model:

$$L_{\text{saturated}}^c = 0. \tag{21}$$

The deviance residual $\Delta_i^c$ is defined as twice the difference between the log-likelihoods of the saturated model and the task net:

$$\Delta_i^c = 2(L_{\text{saturated}}^c - L_i^c) = -2L_i^c. \tag{22}$$

The deviance residual for each representation on a task net is given by:

$$\Delta_i^c = -2\log(\hat{y}_{\mathbf{r}}^c) \tag{23}$$

This shows that the deviance residual for each observation depends solely on the predicted probability of the true class. A larger deviance residual indicates a worse fit for that representation, potentially signaling malicious activity on client $i$'s task net.

## A.2 PROOF OF THEOREM 1

*Proof.* Our detection algorithm sorts clients based on their deviance residuals $\Delta_i$. We aim to determine a threshold $\mathcal{T}$ such that clients with $\Delta_i \leq \mathcal{T}$ are classified as benign, and clients with $\Delta_i > \mathcal{T}$ are classified as malicious. To analyze the misclassification rates, we define the following:

(i) The false positive rate (FPR) is the probability that a benign client is misclassified as malicious:

$$\text{FPR} = \mathbb{P}(\Delta_i > \mathcal{T} \mid i \in \mathcal{B}).$$

(ii) The false negative rate (FNR) is the probability that a malicious client is misclassified as benign:

$$\text{FNR} = \mathbb{P}(\Delta_i \leq \mathcal{T} \mid i \in \mathcal{M}).$$

Our goal is to choose the threshold $\mathcal{T}$ that minimizes the total misclassification rate (TMR), defined as

$$\text{TMR} = \pi_{\mathcal{B}} \cdot \text{FPR} + \pi_{\mathcal{M}} \cdot \text{FNR},$$

where $\pi_{\mathcal{B}} = 1 - \pi_{\mathcal{M}}$ is the proportion of benign clients.

Since we do not assume a specific distribution for the deviation residuals, we apply Chebyshev's inequality to bound the probabilities of misclassification. For any random variable $X$ with expected value $\mu$ and variance $\sigma^2$, Chebyshev's inequality states that for any $\delta > 0$,

$$\mathbb{P}(|X - \mu| \geq \delta) \leq \frac{\sigma^2}{\delta^2}.$$

Applying Chebyshev's inequality to the deviation residuals:

For benign clients:

$$\mathbb{P}(|\Delta_i - \mu_{\mathcal{B}}| \geq \delta \mid i \in \mathcal{B}) \leq \frac{\sigma^2}{\delta^2}.$$

For malicious clients:

$$\mathbb{P}(|\mu_{\mathcal{M}} - \Delta_i| \geq \delta \mid i \in \mathcal{M}) \leq \frac{\sigma^2}{\delta^2}.$$

We choose the threshold $\mathcal{T}$ as the midpoint between the expected residuals of benign and malicious clients:

$$\mathcal{T} = \frac{\mu_{\mathcal{B}} + \mu_{\mathcal{M}}}{2}.$$

This choice sets $\delta = \frac{\mu_{\mathcal{M}} - \mu_{\mathcal{B}}}{2}$. Substituting $\delta$ into the bounds:

$$\text{FPR} \leq \frac{4\sigma^2}{(\mu_{\mathcal{M}} - \mu_{\mathcal{B}})^2},$$

$$\text{FNR} \leq \frac{4\sigma^2}{(\mu_{\mathcal{M}} - \mu_{\mathcal{B}})^2}.$$

Therefore, the total misclassification rate is bounded by

$$\text{TMR} = \pi_{\mathcal{B}} \cdot \text{FPR} + \pi_{\mathcal{M}} \cdot \text{FNR} \leq \frac{4\sigma^2}{(\mu_{\mathcal{M}} - \mu_{\mathcal{B}})^2}.$$

This bound shows that the total misclassification rate decreases as the square of the difference between the mean residuals increases. Specifically, as the separation $\mu_{\mathcal{M}} - \mu_{\mathcal{B}}$ becomes larger relative to the variance $\sigma^2$, the misclassification rate approaches zero.

$\square$

### A.3 BASELINE ATTACKS AND DEFENSES IN EVALUATIONS

Here, we present the baseline attacks and defense algorithms used to evaluate our proposed method.

#### A.3.1 BASELINE ATTACKS

**Min-Max Attack** Shejwalkar & Houmansadr (2021): This model poisoning attack targets FL by crafting malicious gradients that maximize damage to the global model. The attack minimizes the maximum distance between benign and malicious gradients, ensuring that malicious updates remain close to benign ones to evade detection by robust aggregation algorithms while still degrading the model's accuracy.

**Inner Product Manipulation Attack** Xie et al. (2020): This attack compromises Byzantine-tolerant Stochastic Gradient Descent (SGD) algorithms by manipulating the inner product between the true gradient and the aggregated gradient. Adversaries design Byzantine gradients so that the inner product becomes negative, disrupting the descent direction of SGD and leading to a decline in model performance.

**Targeted Label Flipping Attack** Tolpegin et al. (2020): This attack aims to misclassify samples from a specific source class to a target class. Malicious clients flip the labels of samples from the source class to the target class in their local datasets and train their models accordingly, causing the global model to misclassify these samples.

**Edge-Case Backdoor Attack** Wang et al. (2020a): Edge-case backdoors involve altering label data points that are typically correctly classified by the model but are rare or unlikely to appear in regular training or testing data. These backdoors activate only under specific, uncommon conditions, making them hard to detect during standard evaluations. This stealthy approach allows the attacker to maintain the model's performance on typical data while embedding a functional backdoor.

**Distributed Backdoor Attack** Xie et al. (2019): In this attack, the adversary divides the trigger pattern into multiple parts, with each client injecting a partial trigger into a subset of their training samples.

#### A.3.2 BASELINE DEFENSES

**FLShield** Kabir et al. (2024): FLShield is a validation-based defense framework for FL that protects against poisoning attacks. It addresses the validation subject dilemma and the validation integrity

dilemma by generating representative models from local updates. FLShield uses a new metric called Loss Impact Per Class (LIPC) to validate these models and filter out malicious updates, ensuring robust defense against various attacks, including untargeted poisoning, targeted label flipping, and backdoor attacks.

**FedRoLA** Yan et al. (2024): FedRoLA is a robust FL defense algorithm designed to protect against model poisoning attacks through layer-based aggregation. It analyzes the similarity of updates at each layer of a deep neural network using cosine similarity metrics to detect malicious updates. FedRoLA employs a dynamic layer selection and aggregation process that improves threat detection while maintaining model performance.

**FLAIR** Sharma et al. (2023): FLAIR is a defense mechanism for FL systems that protects against model poisoning attacks by detecting abnormal patterns in client gradient updates, focusing on changes in gradient direction (flip-score). It assigns reputation scores to each client based on these patterns and adjusts their contributions to the global model accordingly. FLAIR is effective against advanced untargeted model poisoning attacks, such as directed deviation attacks, and can defend against both white-box and adaptive attacks. However, it relies on a predefined reputation decay parameter, which controls the weight given to past versus current client behavior.

**Trimmed-mean** Yin et al. (2018); Xie et al. (2018): Trimmed-Mean is a robust aggregation algorithm used in FL to defend against malicious or outlier updates. It processes each dimension of the input gradients separately by removing the largest and smallest $\beta$ values from the set of local model updates for each dimension. After removing these extremes, it computes the mean of the remaining $n - 2\beta$ values for that dimension, which becomes the corresponding dimension in the global model update. This method provides resilience against outliers, but its effectiveness depends on choosing the correct $\beta$, which may vary with the proportion of malicious clients.

**Multi-krum** Blanchard et al. (2017): Multi-Krum is a robust aggregation algorithm designed to reduce the impact of malicious updates in FL. Building on the original Krum algorithm, Multi-Krum selects gradients that are closest to their $n - m - 2$ neighbors based on squared Euclidean distance, where $n$ is the number of participating clients and $m$ is an upper bound on the number of malicious clients. Selected gradients are added to a selection set and removed from the pool of gradients. This process continues until a specified number of gradients are selected, which are then averaged to form the global model update.

REPRODUCIBILITY CHECKLIST

1. **For all authors:**

    (a) Do the main claims made in the abstract and introduction accurately reflect the paper's contributions and scope? **Yes**.

    (b) Have you described the limitations of your work? **Yes**.

    (c) Have you read the ethics guidelines and ensured that your paper conforms to them? **Yes**.

2. **If you are including theoretical results:**

    (a) Have you stated all of the assumptions used in your theoretical results? **Yes**.

    (b) Have you provided complete proofs of all theoretical results? **Yes**.

3. **If you ran experiments:**

    (a) Have you included the code, data, and instructions needed to reproduce the main experimental results, including all requirements (e.g., a `requirements.txt` file with explicit versions), an informative `README` with installation and execution commands (either in the supplemental material or as a URL)? **Yes**.

    (b) Have you specified all the training details (e.g., data splits, preprocessing, search spaces, fixed hyperparameter settings, and how they were chosen)? **Yes**.

    (c) Did you ensure that you compared different methods (including your own) on exactly the same benchmarks, including the same datasets, search space, code for training, and hyperparameters for that code? **Yes**.

    (d) Did you run ablation studies to assess the impact of different components of your approach? **Yes**.

    (e) Did you use the same evaluation protocol for the methods being compared? **Yes**.

    (f) Did you perform multiple runs of your experiments and report random seeds? **Yes**.

    (g) Did you report error bars (e.g., with respect to the random seed after running experiments multiple times)? **Yes**.

4. **If you are using existing datasets:**

    (a) Does this paper rely on one or more existing datasets? **Yes**.

    (b) Is a motivation provided for why the experiments are conducted on the selected datasets? **Yes**.

    (c) Are the datasets publicly available? **Yes**.

    (d) Are the datasets accompanied by appropriate citations? **Yes**.

    (e) Have you described any modifications made to the datasets? **Yes**.

5. **If you are introducing new datasets:**

    (a) Have you described the properties of the datasets? **N/A**.

    (b) Have you detailed the data collection process? **N/A**.

    (c) Is the dataset publicly available? **N/A**.

    (d) Have you provided a license for the dataset? **N/A**.

    (e) Have you provided the raw data behind the results (e.g., the input data, the preprocessed data, etc.)? **N/A**.

    (f) Have you reported any potential data biases and discussed how these biases may affect the results? **N/A**.

    (g) Have you documented the consent process for data collection? **N/A**.

    (h) If the dataset is not publicly available, have you provided a justification? **N/A**.

6. **If you used crowdsourcing or conducted research with human subjects:**

    (a) Did you include the full text of instructions given to participants and screenshots? **N/A**.

    (b) Did you describe any potential participant risks, with links to Institutional Review Board (IRB) approvals, if applicable? **N/A**.

    (c) Did you include the estimated hourly wage paid to participants and the total amount spent on participant compensation? **N/A**.

