# OpenReview forum: "When Large Models Meet Generalized Linear Models: Hierarchy Statistical Network for Secure Federated Learning"
_ICLR.cc/2025/Conference — ICLR 2025 Conference Withdrawn Submission_

### Official Review · Reviewer_9SKb · 2024-10-22

**Soundness:** 1
**Presentation:** 2
**Contribution:** 2
**Rating:** 3
**Confidence:** 4

**Summary:**

This work explores the use of federated learning to address the challenges of large model training and proposes a defense method against untargeted poisoning attacks. GLM-based statistical methods are employed to safeguard the fine-tuning process of large pre-trained models by dividing the model into pre-trained and trainable parts. Empirical evaluations are conducted using three image classification datasets, with metrics such as attack success rate used for performance assessment.

**Strengths:**

The topic of decentralized large model training is compelling, especially with consideration of the security of such models.

**Weaknesses:**

However, the related work and background knowledge are not clearly separated from the work's contributions, which include the decentralized fine-tuning framework and the proposed defense mechanism.

**Questions:**

1. The triplet loss seems to be an arbitrary choice for aligning the representations learned by different clients. Other loss functions, such as InfoNCE [1] and MMD loss [2], could also be suitable. Could you compare and discuss the selection of the loss function for federated alignment?
[1] Contrastive Predictive Coding
[2] Feature Distribution Matching for Federated Domain Generalization

2. Beginning from Line 259, the text discusses general knowledge regarding the security of federated learning. Could you clarify the specific threat models and assumptions for this work? Similarly, when addressing the defense method, please provide more details about the attacks considered, including attack frequency and the accessibility of client data.

3. The datasets used in the study are relatively simple. When there are large differences within client data, this variation could be confused with malicious activity. How does the proposed defense handle such situations?

4. More recent attack baselines should be included for comparison on Line 360.

5. The limitations of the work are not clearly explained. For example, what potential tasks could this method be applied to, particularly in the context of text generation? What types of attacks might be encountered in such cases? Additionally, what could be causing the high false positive rates, and what are potential solutions?

---

### Official Review · Reviewer_h6Xd · 2024-10-30

**Soundness:** 3
**Presentation:** 2
**Contribution:** 2
**Rating:** 5
**Confidence:** 3

**Summary:**

The paper presents a novel framework called HStat-Net for integrating generalized linear models (GLMs) with large pre-trained models in the context of FL. This framework aims to make GLMs work more effectively in FL. With the help of the proposed HStat-Net, this paper introduces a mechanism called FEDRACE, which detects poisoning attacks using deviance residuals, a statistical method used in GLMs. Extensive experiments validate that FEDRACE effectively detects various poisoning attacks while delivering outstanding performance compared to SOTA defenses.

**Strengths:**

1. This paper provides a theorem to calculate the upper bound of detection error and adds a theoretical foundation to the detection process, enhancing the credibility of the proposed methods.
2. The paper addresses the scalability and efficiency of the detection process, particularly for mobile devices, by reducing reliance on client-side validation and eliminating the need for pre-defined parameters.

**Weaknesses:**

1. The description of targeted attacks in Lines 246-247 of the paper is inaccurate. The purpose of the targeted attacks is not to reduce the model’s accuracy for the targets.
2. The paper does not explain the specific setting of the attack methods. Thus, it is difficult to ensure that the attacks themself are effective.
3. This paper should show the experimental results of various attacks without deploying any defense. This can better demonstrate the effectiveness of the proposed approach.
4. This paper needs to conduct more experiments with advanced FL backdoor attack and defense methods, such as and Neurotoxin [1], A3FL [2], FLAME [3].

Reference
[1] Zhang, Zhengming, et al. Neurotoxin: Durable backdoors in federated learning. ICML’2022.
[2] Zhang, Hangfan, et al. A3FL: adversarially adaptive backdoor attacks to federated learning. NeurIPS’ 2023.
[3] Nguyen et al. FLAME: Taming backdoors in federated learning.USENIX Security’ 2022.

**Questions:**

Please refer to the weaknesses.

---

### Official Review · Reviewer_iMRp · 2024-11-03

**Soundness:** 2
**Presentation:** 3
**Contribution:** 2
**Rating:** 5
**Confidence:** 3

**Summary:**

This paper introduces a new framework named HStat-Net to integrate GLMs with large pre-trained models, and proposes a new method named FedRACE to detect potential poisoning attacks for downstream tasks in federated learning. It designs a two-step training process and leverages representation refining from the Statistical Net to obtain more discriminative representations, thereby making class-wise representation outlier detection more effective. The experimental results show that FedRACE has an outstanding performance in detecting FL poisoning attacks compared to SOTA methods.

**Strengths:**

1. The paper is well-organized and easy to follow, figures and tables are helpful and easy-to-understand. The background information is introduced in detail, e.g., the evaluation of partially fine-tuned solution for large pre-trained models in federated learning.
2. The proposed HStat-Net has a novel two-step training process that separately refines representation and performs the downstream task, making the poison detection more effective.
3. A theorem along with its proof is provided to support the idea.
4. The experimental results demonstrate superior performance compared to SOTA methods.

**Weaknesses:**

1. I feel confused about the application of GLM theory. As described in Section 3, the HStat-Net integrates large pre-trained models with GLMs. However, from the description in Section 5.1, the HStat-Net simply appends two fully-connected layers to the CLIP/ResNet image encoder. Could the authors further explain the relation between the HStat-Net and the Generalized Linear Model? How does the GLM theory be applied here?
2. As described in Section 3, the statistical net performs dimensionality reduction to enhance privacy. Could the authors provide further clarification on how privacy is specifically improved through this process?
3. The experiments mainly utilize CLIP (ViT-B-32) and ResNet-152 as image encoders. Since the paper researches on large pre-trained models, I suggest the authors conduct supplementary experiments on larger CLIP image encoders in OpenCLIP [1] to thoroughly evaluate the performance and generalization capabilities of the approach across different model scales and embedding dimensions.
4. Table 4 seems to be missing a control group without any defense, e.g., FedAvg.

[1] https://github.com/mlfoundations/open_clip

**Questions:**

Please check the questions in the weaknesses above.
Additionally,
1. Table 1 compares the performance of three training methods. But I failed to find any details of the model configuration, dataset, and hardware setup used for this evaluation. Could the authors supplement the setting of the model, dataset, and hardware devices for this evaluation?
2. Could the authors explain the meaning of '2^2' and '+0.1' in Eq.4 ?
3. Why the output dimension of 256 is selected for the statistical net?

---

### Official Review · Reviewer_rc7K · 2024-11-04

**Soundness:** 3
**Presentation:** 2
**Contribution:** 2
**Rating:** 5
**Confidence:** 3

**Summary:**

This paper proposed a defense method for poisoning attack (targeted, untargeted, and backdoor) in federated learning. Based on a pre-trained model as feature extractor, this paper proposes to train a HStat-Net on top of the feature extractor; the HStat-Net contains two fully connected layers, one layer for representation learning (with dimension reduction), and an additional layer for classification. In addition to the classification loss for learning both layers of HStat-Net, the representation learning layer is also trained with a TripLet loss. In the proposed FedRace method, poisoned clients are detected by looking at whether the local classifiers are “aligned” with the global (robust) representation; a score is generated and all clients are ranked to find a good threshold for detection.

**Strengths:**

HStat-Net, or more generally, representation learning, for poisoning defense makes sense. The proposed method (as far as I can tell with my limited knowledge of poisoning literature) appears to be new. The paper is easy to follow.

**Weaknesses:**

The proposed method is only evaluated on image classification tasks, and the tasks are relatively easy with a small number of classes and total number of samples. In general, it is a bit unclear why HStat-Net is much better than robust aggregation, e.g, https://arxiv.org/abs/1912.1344

**Questions:**

In section 4, is HStat-Net trained together with client detection? I assume it is the case for FedRace, but not entirely sure because of the writing: section 4 follows section 3, and section 3 does not consider detection. Additionally,
- I am not convinced representation learning by HStat-Net (two linear layers) in Section 3 without client detection in Section 4 can be considered a significant contribution. Comparing to a pre-trained model as the feature extractor alone is not convincing. There are significant progress in representation learning in both centralized training and federated learning, I would request the authors to at least discuss the relationship to [neural collapse](https://arxiv.org/abs/2008.08186), [proto learning](https://arxiv.org/abs/2105.00243), and other federated representation learning methods such as [1](https://arxiv.org/abs/2211.10844) [2](https://arxiv.org/pdf/2210.00092)
- Could the authors clarify whether the baseline methods also use a pre-trained model as feature extractor, and how many additional layers do they train/fine-tune? If they are trained from scratch, I would like to see more discussion to justify the comparison.

I may have missed it. Does the adversary know a pre-trained model will be used for feature extractor, and detection-based defense will be deployed?

Assuming global information across clients by ranking in Section 4.3 seems to be a strong requirement, and increasing potential privacy risks.

I may have missed it, could authors discuss the effect of heterogeneity? To what extent client heterogeneity would affect the defense performance? The discussion can be empirical or theoretical.

Minor: the authors might consider the usage difference of \citep and \citet in writing.

---

### Official Review · Reviewer_G6Rx · 2024-11-04

**Soundness:** 3
**Presentation:** 3
**Contribution:** 2
**Rating:** 6
**Confidence:** 3

**Summary:**

This paper introduces a novel approach for enhancing security in federated learning (FL) using large pre-trained models through a framework called the Hierarchy Statistical Network (HStat-Net). The paper integrates generalized linear models (GLMs) with large models to leverage statistical methods in FL settings, particularly to combat poisoning attacks. The proposed HStat-Net combines a pre-trained feature extractor, a statistical net, and a task-specific GLM head, creating an architecture that optimizes for both model accuracy and security. Building on HStat-Net, the authors introduce FEDRACE, a method for detecting poisoning attacks using deviance residuals derived from GLM-based models. Experimental evaluations across datasets (e.g., CIFAR-100, Food-101, Tiny ImageNet) show that FEDRACE outperforms state-of-the-art defenses.

**Strengths:**

1. The introduction of GLMs within the FL framework for secure learning is novel, particularly with HStat-Net facilitating attribute separability, which is important for effective anomaly detection.
2. HStat-Net’s modular architecture enables efficient adaptation of large models in FL, addressing both computation and security concerns.
3. FEDRACE demonstrates high efficacy in identifying various poisoning attacks across different datasets, with performance metrics showing improvements over baseline defenses.
4. Experimental results validate the robustness and generalization of FEDRACE and HStat-Net in a diverse range of FL settings, including non-IID data distributions and variable client participation.

**Weaknesses:**

1. HStat-Net is designed primarily for classification tasks, which may limit its generalizability to other types of tasks, such as text generation or multi-modal learning.
2. While the paper claims HStat-Net is scalable, the additional computational overhead of integrating GLMs and handling large-scale datasets could pose challenges in real-world applications with resource-constrained devices.
3. This work has provided analysis for total misclassification rate, but it lacks theoretical justification for the convergence guarantee of the proposed method without/with attack.

**Questions:**

1. How does HStat-Net perform on tasks beyond classification, and what modifications would be necessary to support broader applications?
2. How does FEDRACE adapt to dynamically changing client data distributions, particularly in highly non-IID scenarios?
3. Can the approach be extended to handle federated learning with limited labeled data, where unsupervised or semi-supervised learning might be more practical?

---

### Official Review · Reviewer_iCpu · 2024-11-07

**Soundness:** 1
**Presentation:** 2
**Contribution:** 1
**Rating:** 3
**Confidence:** 5

**Summary:**

The paper considers the setting of federated anomaly detection in a fine-tuning process, where the host fine-tunes a (large) pre-trained model instead of learning from scratch. The paper argues that Building on that observation, the paper proposes a network called Hierarchy Statistical Network (HStat-Net), which claims to be motivated by GLMs, and used for anomaly detection in federated learning. The network is basically additional layers on top of the pre-trained models, which are then fine-tuned to make the extracted features more discriminative. The paper also proposes a method called FedRACE, based on HStat-Net, to detect poisoning attacks.

**Strengths:**

The methodology described in this paper is easy to understand.

**Weaknesses:**

First thing first, I admit that I am no expert in anomaly detection in federated learning. I would try to comprehend the paper as a general audience. However, I have a very hard time trying to understand the motivation of this paper and why it is special. Here are the reasons:

## Weak motivation and novelty
Though this paper's motivation is repeatedly emphasized as based on GLMs, I see little to no relation between this method and GLMs. The following questions could help motivate the paper much better:
1. What is the theoretical connection between the proposed methods and GLMs?
2. Why would the proposed method be specifically motivated by GLMs? Why don't I design methods motivated by other statistical models for anomaly detection, but GLMs?

As far as I understand, the proposed method just adds additional feature extractors to the pre-trained models. The idea here is very common and well-known in the literature. Moreover, the following questions would also help:
1. Why would we want to add additional feature extractors on top of pre-trained models? Why don't I just keep the last 2 or 3 layers of the pre-trained models unfreeze instead?
2. Which properties make the proposed methods special compared to similar ideas on refining the features of pre-trained models in the literature?

## Weak experimental setups
Since the authors do not mention other anomaly detection literature in the Related works, the authors do it here. It is not just simply mentioning the name/sources of those methods, but also elaborates on the key features of previous methods, and comments on the key differences between the proposed method and the other methods in the literature. Besides, comments on backdoor attack methods used for experiments, and the intuition why this method works well against those attack methods are also helpful.

In Table 4, please highlight the methods that achieve the best performances (make it bold or red ...). This would make the readers keep track of the outstanding methods more easily.

## Superficial Related Works
Section 6 - Related works - is poorly written. The authors should consider adding paragraphs (at least) for the literature on: (1) Federated Learning, (2) Anomaly Detection in Federated Learning, and (3) Large pre-trained models. Besides, there are lots of acronyms that are not defined: GAMs, and LIME.

The writing in that section right now is very uncomprehensive and desultory.

## Other errors/typos
The typos are not carefully checked. Here are some examples:
1. The citations in this paper are quite a mess. Please you ``\citep{(paper)}``, ``\citet{(paper)}`` instead of just ``\cite{(paper)}``.
2. In Equation 2, should it be $g(\mathbb{E}(Y \mid {\color{red} \mathbf{R}})) = \mathbf{R}^\top \mathbf{w}_{h}$?
3. In Equation 3, should it be $\hat{y}_i = \mathbf{h}_i(\mathbf{s}_i({\color{red} \phi(\mathbf{x})})) = \psi_i(\phi(\mathbf{x}))$?
4. Multiple space typos, for example on page 3, line 109, "...generalizationCai et al."

## Conclusion
The above are just uncomprehensive suggestions for this paper. I believe the paper's presentation and soundness could be improved. But as of this state, I cannot recommend an acceptance for this paper.

**Questions:**

See the Weaknesses above.

---

### Note · Authors · 2024-11-25

**Comment:**

Dear Reviewers,

We appreciate the time and effort you have dedicated to evaluating our work.

After carefully considering your comments, we have decided to withdraw our submission from ICLR. We believe that addressing the issues raised will significantly improve the quality and impact of our research. We plan to undertake a more in-depth revision of the manuscript, including clarifying the theoretical connections to generalized linear models (GLMs), enhancing our experimental setup with additional datasets and advanced attack methods, expanding the related works section, and correcting all identified errors and formatting issues.

Once we have thoroughly revised the paper, we intend to resubmit it for consideration in a future conference or journal. Your constructive feedback has been instrumental in guiding our improvements, and we are committed to enhancing the clarity and robustness of our work based on your suggestions.

Thank you again for your support and insightful comments.

Sincerely,

Authors

**Withdrawal Confirmation:**

I have read and agree with the venue's withdrawal policy on behalf of myself and my co-authors.